# Raffinose Ameliorates DSS-Induced Colitis in Mice by Modulating Gut Microbiota and Targeting the Inflammatory *TLR4–MyD88–NF-κB* Signaling Pathway

**DOI:** 10.3390/foods13121849

**Published:** 2024-06-13

**Authors:** Peng Zhang, Yuyang Xue, Zhengyu Cao, Yaya Guo, Xiaotong Pang, Cheng Chen, Wenju Zhang

**Affiliations:** College of Animal Science and Technology, Shihezi University, Shihezi 832000, China; 15297594947@163.com (P.Z.); xueyuyang1998@sina.com (Y.X.); czy19825522981@163.com (Z.C.); 15729930559@163.com (Y.G.); pxt1524896983@sina.com (X.P.); zhangwj1022@sina.com (W.Z.)

**Keywords:** raffinose, colitis, DSS, *TLR4-MyD88-NFκB*, gut microbiota

## Abstract

This study aimed to explore the protective effects of raffinose (Raf) against inflammatory bowel disease in mice with colitis. Mice were administered 100, 200, or 400 mg/kg Raf for 21 d, followed by drinking-water containing 3% dextran sulfate sodium salt (DSS) for 3 d. Thereafter, the phenotype, pathological lesions in the colon, cytokines levels, and gut microbiota were evaluated. Treatment with Raf reduced the severity of the pathological changes in the colon, mitigating the reduction in colon length. Following Raf intervention, serum levels of inflammatory cytokines (IL-2, IL-6, IL-1β, and TNF-α) tended to return to normal. These results suggest that the anti-inflammatory effects of Raf are associated with a reduction in *TLR4–MyD88–NF-κB* pathway expression in mouse colonic tissues. Analysis of gut microbiota abundance and its correlation with colitis parameters revealed that DSS-induced dysbiosis was partially mitigated by Raf. In conclusion, Raf exerts a protective effect in colitis by modulating the gut microbiota and *TLR4-MyD88-NF-κB* pathway.

## 1. Introduction

Inflammatory bowel disease (IBD), Crohn’s disease, and ulcerative colitis (UC) are characterized by chronic inflammation of the gastrointestinal tract [1]. These conditions significantly increase the risk of colorectal cancer (CRC) in developed regions [2]. IBD affects over two million individuals in the United States, with estimates suggesting that this could increase to approximately four million by 2030 [3]. The prevalence is high in Europe (at 505 per 100,000), in Canada (at 248 per 100,000) [4,5], and in the United States (at 21.4 per 100,000). A significant increase in IBD cases has been observed in Asian countries, with a higher incidence among females and a demographic shift towards younger populations. This trend is largely attributed to substantial changes in lifestyle and environmental factors; these changes are believed to interact with genetic predispositions [6].

UC, a chronic inflammatory bowel disease affecting the rectum and colon, exhibits unpredictable severity. It is often associated with a genetic predisposition that may be activated by environmental triggers. It is characterized by a weakened intestinal barrier, alterations in the gut microbiota, and a suboptimal immune response [7]. Sustained inflammation may lead to irreversible damage to the intestinal mucosal barrier [8]. The recurrent nature of UC and its resistance to complete cure present significant clinical challenges. The current therapeutic options encompass a range of pharmacological agents and advanced strategies, including stem cell therapies [9]. However, these treatments do not achieve a cure and are primarily aimed at managing symptoms. Consequently, there is an urgent need to develop more effective and potent medications to better address the needs of patients with UC.

The development of UC is likely influenced by a complex interplay of genetic factors and environmental interactions [10]. The gut microbiota, which is susceptible to modulation by environmental factors, is integral to immune function and represents a significant target for managing inflammation [11]. Extensive research has demonstrated a link between UC and a disrupted gut microbiota [12]. The microbial community within the gastrointestinal tract of individuals with UC exhibits a compositional and functional shift that diverges from the norm observed in healthy individuals [13]. The intestinal ecosystem of those afflicted by UC is characterized by reduced microbial diversity and a decline in beneficial probiotic bacteria, such as *Bifidobacterium*, *Lactobacillus*, and *Faecalibacterium prausnitzii*. There patients exhibit a notable increase in inflammatory bacteria, particularly *Enterobacteriaceae*, in stark contrast to the gut microbiota of healthy individuals [14].

The oligosaccharide Raffinose (Raf), composed of D-galactose, D-glucose, and D-fructose residues, cannot be broken down in the stomach or small intestine of monogastric animals. However, the fermentation of Raf in the large intestine can provide nutrition for gut bacteria. As a result, Raf is considered a prebiotic [15]. As Raf can be produced and refined from the waste liquid from cottonseed meal production, it has promising environmental and economic value. Raf increases the relative abundance of *Bifidobacterium* [16] and *Lactobacillus* [17] species and the production of short-chain fatty acids and CO_2_ [18]. It enhances the fermentation of *Enterococcus faecium* and *Streptococcus pneumoniae*, while reducing the abundance of *Proteobacteria*, including *Escherichia coli*, in vitro [18]. Consequently, the consumption of fermentable oligosaccharides, disaccharides, monosaccharides, and polyols may benefit those with IBD symptoms.

Raf can disrupt the adhesion of pathogens to host cells by mimicking cell receptors [19] and can increase the abundance of probiotic bacteria, including *Lactobacillus*, *Akkermansia*, and *Pseudoflavonifractor* [20]. It is therefore considered a potential prebiotic oligosaccharide. In hybrid sturgeon, dietary supplementation with Raf improved growth performance, probably improving intestinal morphology, altering intestinal microbial composition, and enhancing immunity [21]. In a broiler chicken model, Raf in ovo injection modified ileum mucosa morphology and improved growth performance and immunity [22]. Raf and stachyose increase Fe bioavailability and brush-border membrane functionality [21]. In broilers, dietary oligosaccharides, especially stachyose, improve gut fermentation and reduce fecal indole levels [20]. The odor compound reducing ability of Raf is lower than that of stachyose but better than that of soybean oligosaccharides [20]. In growing pigs, dietary Raf (at 0.2% and 0.5%) partially improved intestinal morphology and mucosal barrier function; reducing both feed intake and nutrient digestibility reduced growth performance and induced a humoral immune response [23].

Overall, Raf and its family member stachyose improve intestinal structure, immunity, and gut microbiota composition. Raf occurs naturally in soybeans, beets, cotton seeds, cabbage, and various vegetables [24]. However, the use of Raf as a nutritive prebiotic has received little attention. Therefore, in this study, DSS-induced colitis model mice were used to explore the potential functions of Raf, including improving gut inflammation, restoring intestinal morphology, and altering intestinal microbial composition. We hope that these findings will provide dietary guidance for patients with IBD and guide the development new strategies for relieving drug-related side effects.

## 2. Materials and Methods

### 2.1. Animals and Treatments

Raf (>99% purity) was acquired from Zhongtang Ruide Biotechnology Co., Ltd. (Beijing, China), and dextran sulfate sodium salt (DSS, 36–50 kDa, >98% purity) from Kulai Bo Technology Co. (Beijing, China).

Five-week-old male Institute of Cancer Research mice (ICR) (21 ± 2.1 g) were obtained from Pengyue Experimental Animal Breeding (Jinan, China). The 40 mice were kept in specific pathogen-free conditions with a temperature of 22 ± 2 °C, relative humidity of 55–60%, and a consistent 12 h light/12 h dark cycle. Before the experiment, all mice were acclimatized to the environment for 5 d, then randomly divided into a control group (CON), DSS group (3% DSS), low-dose Raf+DSS group (LRaf+DSS), medium-dose group (MRaf+DSS), and high-dose Raf+DSS group (HRaf+DSS), with eight mice in each group. Normal saline was administered to the control mice for 24 d. To induce colitis in the model group, 3% DSS solution was administered for 3 d (Figure 1). The LRaf+DSS received oral Raf daily at 100 mg/kg body weight for 21 d, followed by 3% DSS for 3 d. The MRaf+DSS group was administered oral Raf at 200 mg/kg daily for 21 d followed by 3% DSS for 3 d. The HRaf+DSS received oral Raf at 400 mg/kg daiky for 21 d, followed by 3% DSS. During the experiment, all mice had unlimited access to water and food. Weight loss, stool consistency, and gross bleeding were recorded daily to assess the disease activity index (DAI) [25]. All animal experiments adhered to the guidelines of the Bioethics Committee of Shihezi University in accordance with animal protocol (approval number A2024-206).

### 2.2. Sample Collection

Mouse blood samples were collected via the eyeball enucleation method and placed in 1.5 mL Eppendorf (EP) tubes. The tubes were subsequently stored at 4 °C for 2 h to allow settling. The samples were centrifuged at 3500 rpm for 15 min to separate the supernatants. The supernatant was reserved and stored at −20 °C for further analysis, including of biomarkers of oxidative stress, antioxidants levels, and inflammatory and immune responses. Following blood collection, the mice were euthanized by cervical dislocation. Colon length and thymus and spleen weight were measured to assess the extent of inflammation and damage in the DSS-induced colitis model. The colon mucosae were scraped onto glass slides to extract total RNA, which was immediately reverse transcribed into cDNA for quantitative real-time (qRT)-PCR analysis.

### 2.3. Determining Thymus and Spleen Indices

At the end of the experiment, the mice were euthanized via cervical dislocation. The spleens and thymuses were completely removed and carefully weighed to calculate the organ indices, as follows:$$Spleen\;index\left(mg/10\;g\right)=\frac{spleen\;weight\left(mg\right)}{body\;weight\left(g\right)}\times 10$$
$$Thymus\;index\;(mg/10\;g)=\frac{thymus\;weight\;(mg)}{body\;weight\;(g)}\times 10$$

### 2.4. Histological Analysis

Fresh colonic tissues were fixed in 4% paraformaldehyde for 24 h at room temperature and embedded in paraffin. Subsequently, the intestinal tissues were cut into 5 μm thick slices, stained with hematoxylin and eosin (H&E), and photographed for histopathological aberrations using a Pannoramic MIDI digital section scanner (3DHISTEC, Budapest, Hungary). The tissue sections were scored according to Zhang’s criteria [26].

### 2.5. Determination of Serum Cytokines Levels

Serum concentrations of cytokines, including interleukin (IL)-2, IL-6, IL-1β, and tumor necrosis factor-alpha (TNF-α), were measured using ELISA kits (Enzyme Linked Biotechnology, Shanghai, China), following the manufacturer’s instructions.

### 2.6. Determination of Serum Antioxidant Capacity in Serum

Serum concentrations of various biomarkers, including superoxide dismutase (SOD), glutathione peroxidase (GSH-Px), catalase (CAT), and malondialdehyde (MDA), were quantified using ELISA kits (Enzyme Linked Biotechnology), following the manufacturer’s protocols.

### 2.7. Determination of Immunityin Serum

Serum concentrations of immunoglobulin, including immunoglobulin (IG)-sIgA, IgG, and IgM, were measured using ELISA kits (Enzyme Linked Biotechnology), following the manufacturer’s instructions.

### 2.8. qRT-PCR Analysis

The colonic mucosa, one of the largest immune organs in the digestive tract, plays a crucial role in maintaining inflammation and immune function in mice. Toll-like receptors (including *TLR4*) are important immune receptor that can recognize bacterial endotoxins and trigger inflammatory responses. Myeloid differentiation primary response gene 88 (*MyD88*) is a key protein in the *TLR* signaling pathway. *NF-κB* is a transcription factor that regulates the expression of inflammatory genes, while *TNF-α* and *IL-6* are inflammatory mediators involved in regulating inflammatory responses. Total RNA was extracted from the colonic mucosa using a TransZol Up Plus RNA Kit (TransGen, Beijing, China). The concentration and integrity of RNA were measured using a 0.2 µL dose using a Nanodrop 2000 spectrophotometer (Thermo Fisher Scientific, Waltham, MA, USA). cDNA was synthesized from RNA via reverse transcription using the HIFIScript cDNA Synthesis Kit. qRT-PCR was conducted using perfectStart Green qPCR SuperMix on a Roche Light Cycler 96, following the operating instructions. All samples were analyzed in triplicate, and relative expression was determined using the 2^−ΔΔCt^ method. Primer sequences targeting the mouse genes *TLR4*, *MyD88*, *NF-κB*, *TNF-α*, and *IL-6* were used, with *β-actin* serving as the reference gene (Table 1).

### 2.9. Gut Microbiota Analysis

Genomic DNA was extracted from each stool sample using the TIANamp Stool DNA Kit (Tiangen Biotech, Beijing, China), following the manufacturer’s instructions. DNA concentration and quality were assessed using a Nanodrop spectrophotometer (Thermo Fisher Scientific) and confirmed by 1.5% agarose gel electrophoresis. The V3 and V4 regions of the 16S rRNA gene were amplified with barcoded primers (338F and 806R) using diluted DNA (1.0 ng/mL) as a template with the Phusion High-Fidelity PCR Master Mix (New England Biolabs, Ipswich, MA, USA). The PCR process involved 30 cycles, each comprising four steps: pre-denaturation at 98 °C for 5 min, denaturation at 98 °C for 30 s, annealing at 56 °C for 30 s, and extension at 72 °C for 30 s. The PCR products were sequenced on an Illumina MiSeq platform (Illumina, San Diego, CA, USA) immediately after purification. An amplification and sequencing of the 16S rRNA V3-V4 region was performed using Majorbio Biopharm Technology (Shanghai, China). Raw sequences were submitted to the NCBI for for Biotechnology Information Sequence Read Archive (accession number PRJNA1100313).

### 2.10. Statistical Analysis

Statistical analysis was conducted using SPSS 16.0. Differences between groups were analyzed using one-way ANOVA, Differences with *p* value of <0.05 were considered significant. Values are shown as the mean ± SEM.

## 3. Results

### 3.1. Raf Alleviated DSS-Induced Colitis Symptoms in Mice

Table 2 presents the effects of Raf on final weight and daily weight gain, as well as the thymus and spleen indices. Relative to the control, the DSS group exhibited a significantly lower body weight and daily weight gain, indicating that DSS induced colitis and potentially led to weight loss and lower daily weight gain. Raf treatment mitigated the DSS-induced weight loss and lower daily weight gain. The three Raf-treatment groups exhibited dose-dependent improvements in body weight and daily weight gain. This suggests that Raf protects against DSS-induced weight loss rescues and daily weight gain. The thymus and spleen indices partially reflect the state of animal’s immune system. Here, Raf treatment exerted a significant effect on the thymus index (*p* < 0.01), particularly in the MRaf+DSS and HRaf+DSS groups. The highest thymus index (29.91 ± 2.33, *p* < 0.01) was observed in the MRaf+DSS group (Table 2). Similarly, the spleen indices were DSS group exhibited a significantly lower in the MRaf+DSS and HRaf+DSS groups than in the DSS group (*p* < 0.01). The highest spleen index (29.15 ± 1.95, *p* < 0.01) was recorded in the HRaf+DSS group. Raf significantly reduced the DAI scores, and the effect in the HRaf+DSS group was the best (*p* < 0.01) (Figure 2A). DSS-induced colitis symptoms, in terms of changes in body weight, daily weight gain, DAI scores, and thymus and spleen indices, were alleviated by Raf in a dose-dependent manner.

### 3.2. Effects of Raf on Colon Histology Examinations in ICR Mice

DSS caused major pathological changes, including shortened colon length, intensified hyperemia and edema, thickening of the intestinal wall, and ulcers (Figure 3A). The reduction in colon length was significantly mitigated by Raf pretreatment (Figure 3B). H&E staining of colonic tissue (Figure 3C) revealed significant crypt defects and widespread inflammation in the colon in the DSS-induced model mice, and Raf successfully protected against this DSS-induced damage. Therefore, Raf may prevent the DSS-induced loss of the intestinal epithelial barrier.

### 3.3. Effects of Raf on the Levels of Antioxidant Factors and Immunocorrelative Cytokine Levels

Relative to the control, DSS gavage significantly reduced serum levels of well-known antioxidant enzymes, including CAT, GSH-Px, and SOD (*p* < 0.01) (Figure 4A–C). The activity of these enzymes was significantly higher following Raf pretreatment, and the increase was dose dependent. Serum levels of MDA, a biomarker of oxidative damage, were significantly elevated after DSS gavage but were significantly lower after Raf pretreatment (using 100–400 mg/kg) (Figure 4D).

Serum levels of IgG, IgM, and sIgA were significantly lower in the DSS group than those in the control (*p* < 0.01) (Figure 4E–G). A 400 mg/kg dose of Raf significantly increased serum IgG, IgM, and sIgA in DSS-treated mice by more than 37%, 63%, and 45%, respectively (*p* < 0.01).

### 3.4. Effects of Raf on Inflammatory Factor Levels

Serum levels of IL-1β, TNF-α, and IL-6 were significantly higher in the DSS group than the in control (*p* < 0.01) (Figure 5A–C). IL-2 levels were significantly lower in the DSS group than in the control (*p* < 0.01) (Figure 5D). Serum IL-1β, TNF-α, IL-6, and IL-2 levels were not significantly different between the LRaf+DSS group and the DSS group (*p* > 0.05).

### 3.5. Effects of Raf on NF-κB Signaling Pathway-Related Gene Expression

The model group exhibited significantly higher mRNA levels of *TLR4*, *MyD88*, *NF-κB*, *IL-6*, and *TNF-α* than the control (*p* < 0.01) (Figure 6A–E). The inhibitory effects of Raf on the levels of these mRNAs were stronger in the MRaf+DSS group than in the LRaf+DSS and HRaf+DSS groups.

### 3.6. Intestinal Microbiota Analysis via High-Throughput Sequencing

The number of OTUs was lowest in the DSS group, with the five groups sharing 1298 intestinal microbiotic OTUs. The control and MRaf+DSS groups shared significantly more OTUs than other groups. Bray–Curtis distance-based PCoA analysis revealed a distinct separation between the control and DSS-treated groups (*R*^2^ = 0.3542, *p* = 0.064) as well as between the DSS and Raf groups (*R*^2^ = 0.4479, *p* = 0.001) (Figure 7B,C). Among the three Raf-treatment groups, only the MRaf+DSS group exhibited significantly higher Chao and Shannon indices than the DSS group (Figure 7D,E), indicating that the MRaf treatment group had greater species richness and increased diversity.

The community level bar chart illustrates the dominant species and their relative abundances at each taxonomic level in each sample (Figure 8). Community analysis revealed that *Firmicutes*, *Bacteroidetes*, and *Desulfobacterota* were the most abundant phyla across all of the sample groups (Figure 8G). Relative to the control, the DSS-induced colitis model mice exhibited significantly higher abundances of *Actinobacteriota*, *Firmicutes*, and *Patescibacteria* (Figure 8A–C), along with lower abundances of *Bacteroidetes*, *Campilobacterota*, and *Desulfobacterota* (Figure 8D–F). Raf supplementation restored the composition of the gut microbiota almost to that of the control.

At the genus level, ten major genera were identified, including *Lactobacillus*, *norank_f_Muribaculaceae*, *Lachnospiraceae_NK4A136_group*, *norank_f_Desulfovibrionaceae*, and *unclassified_f_Lachnospiraceae* as the most abundant taxa across all groups (Figure 9G). Relative to the control, the DSS-induced colitis models exhibited lower abundances of *Parabacteroides*, *PrevotellaceaeUCG-001* and *Oscillibacter* (Figure 9A–C) but higher abundances of *Jeotgaliciccus*, *Enterorhabdus*, and *Alloprevotella* (Figure 9D–F). To identify significant differences in bacterial taxa across all groups, LEfSe analysis was performed using linear discriminant analysis (LDA) scores. Relative to the other groups, the DSS group exhibited significantly higher abundances of *Enterorhabdus* and *Gemella*, the MRaf+DSS group exhibited higher *Prevotellaceae_UCG-001* abundance, and the HRaf+DSS group exhibited higher *Alloprevotella* and *GCA-900066575* abundances (LDA threshold > 3.5) (Figure 9H).

Heatmap analysis revealed the genus-level microbial community composition (Figure 10A). DSS treatment caused higher relative abundances of strains, such as *Helicobacter* (*p* = 0.237) and *Escherichia-Shigella* (*p* = 0.672), and significantly higher *Corynebacterium* abundance (*p* = 0.004). These increases were mitigated in all three Raf-treated groups. Three genera, *Faecalibaculum* (*p* = 0.177), *Roseburia* (*p* = 0.156), and *Ruminococcus* (*p* = 0.612), exhibited lower abundances under the influence of DSS. *Prevotellaceae_UCG-001* was significantly lower under DSS treatment (*p* = 0.019). The three Raf concentrations alleviated the DSS-induced reductions in the abundance of these four beneficial genera to varying degrees. As anticipated, DSS enhanced the relative abundances of *Helicobacter*, *Escherichia-Shigella*, and *Corynebacterium*, and Raf treatments inhibited the growth of these three genera.

Figure 10B illustrates the relationships between the 21 most abundant gut bacteria genera and oxidative stress, inflammation, and immune-related cytokines levels. *Alistipes* and *Bacteroides* abundances significantly affected most of the measured factors. *Alistipes* exhibited a significant negative correlation with levels of the inflammatory markers TNF-α, IL-6, and IL-1β (*p* < 0.01) and a positive correlation with those of sIgA, IgG, IgM, CAT, and GSH-Px (*p* < 0.05). This suggests that increasing *Alistipes* abundance may help to reduce disease activity. In contrast, *Bacteroides* abundance was significantly negatively correlated with levels of TNF-α and of the oxidative stress marker MDA (*p* < 0.05) and was positively correlated with those of IL-2, sIgA, IgG, SOD, CAT, and GSH-Px (*p* < 0.05). The overall abundance of *Colidextribacter* was significantly correlated with levels of sIgA, IgG, IgM, and SOD (*p* < 0.05). The overall abundance of *Candidatus_Saccharimonas* was significantly positively correlated with TNF-α levels and negatively correlated with those of SOD, CAT, and GSH-Px (*p* < 0.01). The overall abundance of *Lactobacillus* was significantly positively correlated with MDA levels (*p* < 0.01) and negatively correlated with sIgA and SOD (*p* < 0.05), indicating opposite trends in their relationships. The overall abundance of *Streptococcus* was significantly negatively correlated with IL-2 and GSH-Px levels (*p* < 0.05).

## 4. Discussion

Symptoms of IBD, including abdominal pain, diarrhea, fever, vomiting, the presence of blood in the stool, anemia, and weight loss, significantly impair quality of life and health, potentially leading to substantial economic strain [27]. Clinically, IBD is managed primarily using conventional therapies such as corticosteroids and immunomodulators, along with biological treatments [28]. However, long-term therapy can lead to numerous side effects, high recurrence rates, and significant cost. Consequently, there is growing interest in exploring alternative treatments. These alternatives, including functional foods and dietary supplements from natural sources, promise fewer adverse effects.

Raf is a naturally occurring oligosaccharide that occurs widely in nature and is used as a medicine, food, and cosmetic additive [15]. As a non-digestible carbohydrate, it cannot be digested and absorbed in the small intestines of humans and animals. Raf plays a pivotal role in maintaining gut health, as it reacts with the large intestine and functions as a prebiotic. This activity fosters the proliferation of advantageous bacterial populations, notably *Bifidobacteria* [16] and *Lactobacillus* [17], thus enhancing the gastrointestinal wellness and immune response of the host. Raf exhibits unique biological activities, including antitumor, anti-hepatitis, anti-cardiovascular, and anti-aging effects.

Given the unique biological activities of Raf, we investigated whether Raf could alleviate DSS-induced colitis by regulating the gut microbiota and inflammatory signaling pathways. In the present study, Raf was also shown to be effective against UC. This included body weight loss, shortened colon, and colonic inflammation; the results are consistent with previous study [29]. All these features were the typical symptoms of a colitis patient [30]; in contrast, Raf treatment significantly improved body weight and rescued the loss in daily weight gain loss in DSS model mice. Histologically, Raf significantly alleviated the pathological damage caused by colitis, achieving a relatively intact colonic structure without evident erosion and ulcers. The DSS-induced colitis group exhibited shortened colon length (Figure 3A), a common indicator of inflammation [31]. Raf significantly inhibited colon shortening, providing important evidence for its anti-inflammatory effects. Therefore, the macroscopic indicators are consistent with our histological findings. We then examined how Raf mitigates colitis via the intestinal microbiota and related inflammatory pathways.

Next, we investigated the effect of Raf on pro-inflammatory factors. Clinically, inhibition of the inflammatory response can help to improve symptoms [32]. DSS triggers UC by increasing the secretion of pro-inflammatory cytokines. Excessive production of pro-inflammatory cytokines (IL-6, IL-1β, and TNF-α) can lead to immune-mediated diseases and cause inflammation. IL-2, a T-cell growth factor, is an anti-inflammatory cytokine that controls T-cell proliferation and differentiation [33]. IL-2 therapy stimulates the immune response, increasing the body’s T-cell count, thereby promoting intestinal development and mucosal barrier function [34,35]. IL-1β has the potential to downregulate occludin mRNA levels, thereby enhancing intestinal tight junction (TJ) permeability [36]. TNF-α acts as an important mediator of inflammation, exacerbating it by causing degradation of the mucosa [37]. IL-6 augments intestinal TJ permeability by activating the c-Jun N-terminal kinase (JNK) signaling pathway [38].

Here, we evaluated the effects of Raf on inflammation using a DSS-induced colitis mouse model, focusing on serum levels of IL-2, IL-1β, TNF-α, and IL-6. Raf treatment resulted in dose-dependent reductions in the levels of pro-inflammatory cytokines IL-1β, TNF-α, and IL-6, particularly in the HRaf+DSS group. There findings suggest a potential dose-dependent anti-inflammatory effect of Raf. Concurrently, the levels of the anti-inflammatory cytokine IL-2 increased in a dose-dependent manner and were highest in the HRaf+DSS group.

These findings indicate that Raf can modulate both pro-inflammatory and anti-inflammatory cytokines in mice; this ability may contribute to the suppression of DSS-induced colitis. The pathogenesis of IBD is associated with excessive and dysregulated immune responses within the intestinal mucosa [39]. DSS impairs the intestinal epithelial barrier, allowing pathogenic bacteria to breach the mucus layer, infect epithelial cells, and induce inflammation [40]. Many studies have demonstrated that the transcription factor *NF-κB* regulates genes linked to inflammation, including *TNF-α*, *IL-1β*, and *IL-6* [41,42]. *TNF-α* and *IL-6,* as pivotal pro-inflammatory cytokines, are integral to the development of inflammation and the pathogenesis of IBD. Recent evidence suggests that the *TLR4-MyD88-NF-κB* signaling pathway is the principal regulator of the inflammatory response [43]. *TLRs* are known to initiate inflammatory responses by engaging the innate immune system [44]. Patients with UC exhibit increased intestinal permeability, facilitating the translocation of toxic substances and pathogens across the intestinal barrier and triggering *TLR4* activation [40]. *TLR4*, an essential innate immune receptor, modulates numerous inflammatory cytokines in IBD [41]. *TLR4*-deficient mice were protected from DSS-induced colitis [43]. Under normal conditions, *NF-κB* is complexed with the inhibitor of kappa B (*IκB*), maintaining it in an inactive state. Via the *MyD88*-dependent pathway, *NF-κB* can be rapidly activated, enhancing the transcription of downstream inflammation-related genes [44].

Here, pretreatment with Raf significantly reduced levels of *TLR4* and *MyD88*, thus inhibiting *NF-κB* activation. Levels of *IL-6* and *TNF-α*, inflammatory mediators downstream of *NF-κB*, were substantially lower in the Raf-treated groups. These reductions imply that Raf may mitigate inflammation by suppressing the transcriptional activity of *NF-κB*. These findings reinforce the regulatory role of Raf in inflammatory responses. Collectively, these findings suggests that Raf exerts anti-inflammatory properties by modulating the *TLR4-MyD88-NF-κB* signaling pathway. This provides a scientific rationale for the potential therapeutic application of Raf in inflammatory diseases.

Gut microbiota play an important role in human health, potentially regulating body inflammation, controlling appetite, and crucially supporting intestinal integrity and function [45]. Dysbiosis of the gut microbiota may be related to the pathophysiological status of the host and may be an important indicator [29]. Patients with IBD exhibit reduced microbial diversity within the inflamed colonic tissues [46]. Consistent with previous findings [32,47], we observed a diminished alpha diversity in DSS-treated mice, based on the Shannon and Chao indices. Raf administration enhanced intestinal microbiota diversity, facilitating distinct clustering of microbiota among the groups, based on PCoA. Therefore, the diversity and quantity of microbiota at the phylum and genus level in the gut can serve as indicators for evaluating gut health. At the phylum level, DSS treatment significantly increased the abundance of *Actinobacteriota*, *Firmicutes*, and *Patescibacteria*. *Bacteroidota* relative abundance is substantially reduced both in patients with IBD and in DSS-induced colitis murine models [48,49]. Here, DSS induced a similar reduction in *Bacteroidota* at the phylum level. DSS treatment increased the relative abundance of *Proteobacteria*, consistent with previous findings [32,50]. The proliferation of *Proteobacteria*, indicative of a disrupted microbiota, is associated with epithelial dysfunction [51]. *Proteobacteria* induce inflammation and modify the gut microbiota, potentially contributing to the progression of IBD [52]. Supplementation with Raf restored the gut microbiota structure to almost the levels observed in the control. At the genus level, oral Raf significantly reduced the abundance of *Jeotgalicoccus* and *Enterorhabdus* and significantly increased that of *Prevotellaceae_UCG-001* and *Alistipes*. *Jeotgalicoccus* is a pathogen typically associated with inflammatory diseases [53]. The abundance of *Enterorhabdus* is positively associated with intestinal inflammation [54]. *Prevotellaceae_UCG-001*, a probiotic with the potential to reduce blood sugar, possesses polysaccharide hydrolases associated with the production of short-chain fatty acids and alleviates glucose and lipid metabolism disorders by stimulating the *AMPK* signaling pathway [54]. The role of *Alistipes* in gut inflammation remains debatable. *Alistipes* is widely suggested to confer a protective effect against colitis [55,56], and *Alistipes finegoldii* treatment has been shown to ameliorate DSS-induced colitis in mice [57].

We further investigated whether gut microbiotic abundance was associated with colitis parameters. *Alistipes* abundance was highly negatively associated with inflammatory cytokine levels, consistent with previous findings [55,58]. Raf corrected gut microbiota imbalances, suppressed pathogenic bacterial growth, and boosted the abundance of potential probiotics. Figure 11 mainly presents the experimental content and the pathway map. However, the mechanisms whereby probiotics regulate the intestinal flora remain unclear. Metabolomic and metagenomic research is necessary to clarify these relationships and identify the specific bioactive compounds involved. 

## 5. Conclusions

In conclusion, supplementation with 200 mg/kg Raf regulated the intestinal immune response and intestinal morphology while improving the gut microbiota imbalance in DSS-induced colitis mice. Levels of pro-inflammatory factors *TNF-α* and *IL-6* were reduced and inhibited via the *TLR4–MyD88–NF-κB* pathway. Based on our findings, the Raf is a promising prebiotic for improving animal intestinal health.

## Figures and Tables

**Figure 1 foods-13-01849-f001:**
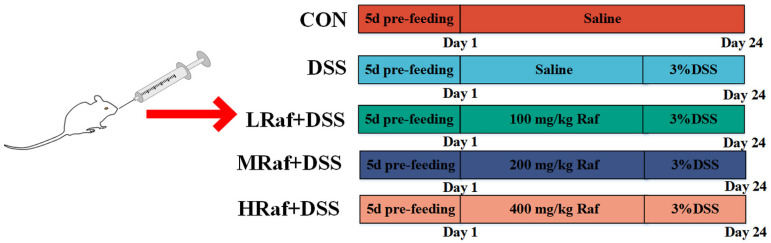
Schematic of the animal experiment design.

**Figure 2 foods-13-01849-f002:**
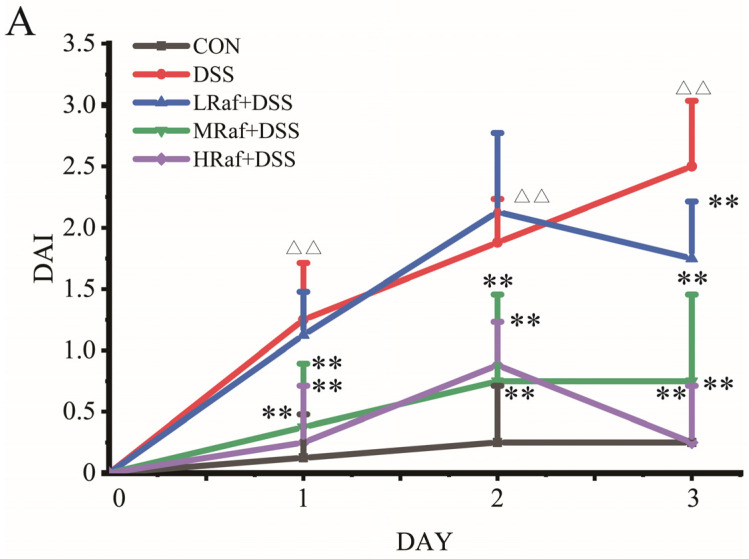
Raffinose (Raf) rescued changes in the physiological parameters of colitis model mice. (**A**) Disease Activity Index (DAI) scores. ^ΔΔ^ *p* < 0.01 relative to the control. ** *p* < 0.01 relative to the DSS group.

**Figure 3 foods-13-01849-f003:**
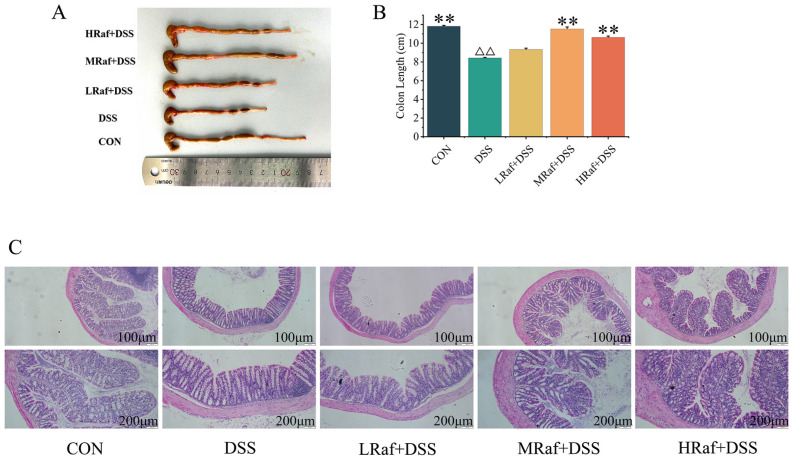
Effects of Raf on the histopathology of a DSS-induced colitis model mouse colon. (**A**) Macroscopic appearance. (**B**) Colon length. (**C**) H&E-stained colon. ^ΔΔ^ *p* < 0.01 relative to the control. ** *p* < 0.01 relative to the DSS group. CON: control; Raf: Raffinose; DSS: dextran sulfate sodium salt.

**Figure 4 foods-13-01849-f004:**
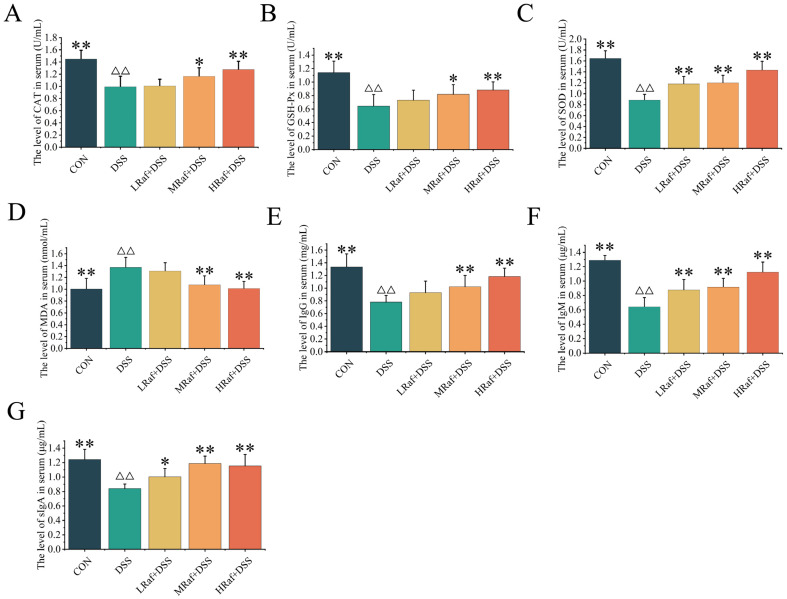
Effects of Raf on antioxidant factor and immunocorrelated cytokine levels. (**A**) CAT, (**B**) GSH-Px, (**C**) SOD, (**D**) MDA, (**E**) IgG, (**F**) IgM, and (**G**) sIgA. ^ΔΔ^
*p* < 0.01 relative to the control. * *p* < 0.05, ** *p* < 0.01 relative to the DSS group. CON: control; Raf: Raffinose; DSS: dextran sulfate sodium salt.

**Figure 5 foods-13-01849-f005:**
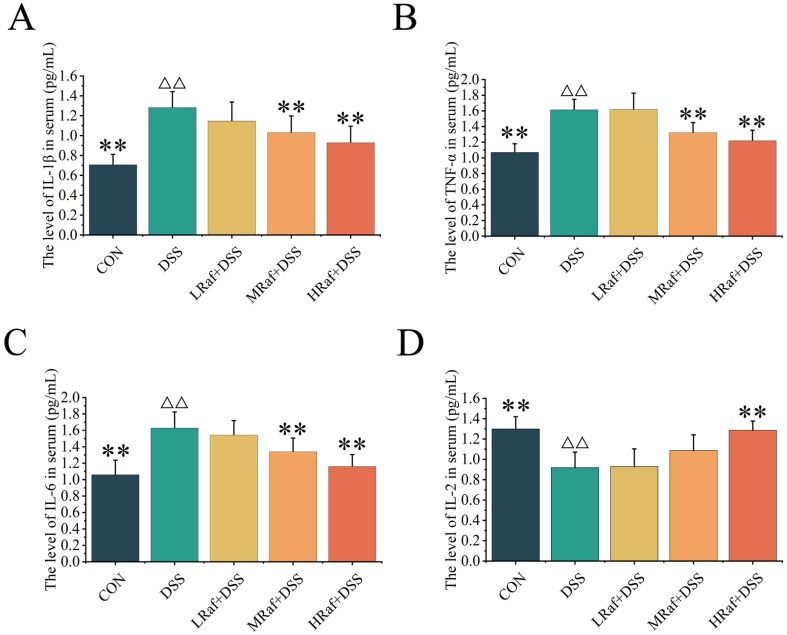
Effects of Raf on inflammatory cytokines. (**A**) IL-1β, (**B**) TNF-α, (**C**) IL-6, and (**D**) IL-2. ^ΔΔ^
*p* < 0.01 relative to the control. ** *p* < 0.01 relative to the DSS group. CON: control; Raf: Raffinose; DSS: dextran sulfate sodium salt.

**Figure 6 foods-13-01849-f006:**
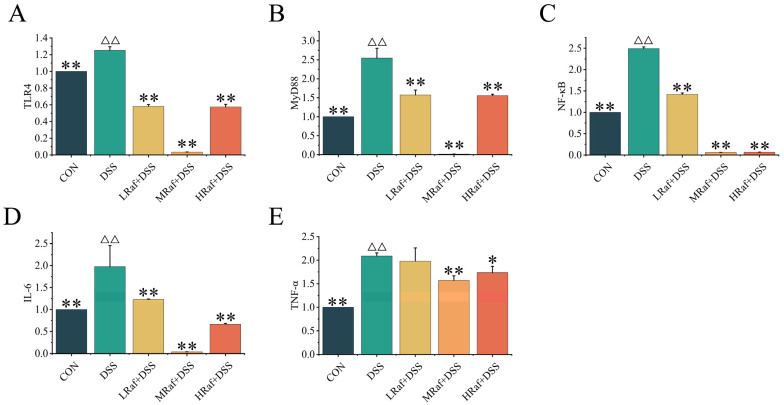
Raf regulated *TLR4–MyD888–NF-κB* signaling pathway components. (**A**) *TLR4* mRNA, (**B**) *MyD88* mRNA, (**C**) *NF-κB* mRNA, (**D**) *IL-6* mRNA, and (**E**) *TNF-α* at the mRNA level. ^ΔΔ^
*p* < 0.01, relative to the control. * *p* < 0.05, ** *p* < 0.01, relative to the DSS group. CON: control; Raf: Raffinose; DSS: dextran sulfate sodium salt.

**Figure 7 foods-13-01849-f007:**
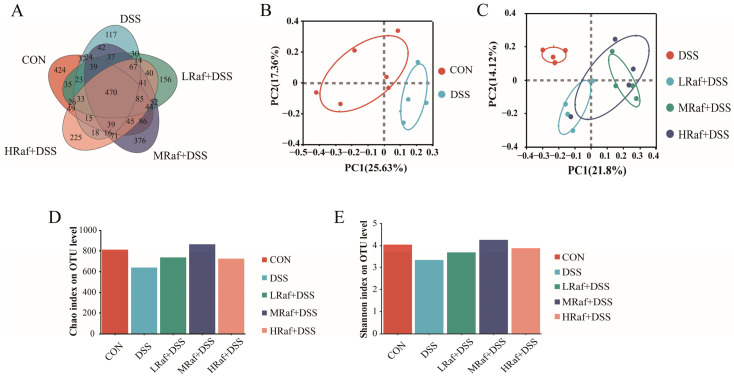
The mice exhibited a rich gut microbiotic diversity. (**A**) Venn diagram showing the overlap of the OTUs in the five groups. (**B**) PCoA plots illustrating the preventative effect of Raf on colitis. (**C**) PCoA plots illustrating the preventative effect of Raf. (**D**) Shannon index. (**E**) Chao index. CON: control; Raf: Raffinose; DSS: dextran sulfate sodium salt.

**Figure 8 foods-13-01849-f008:**
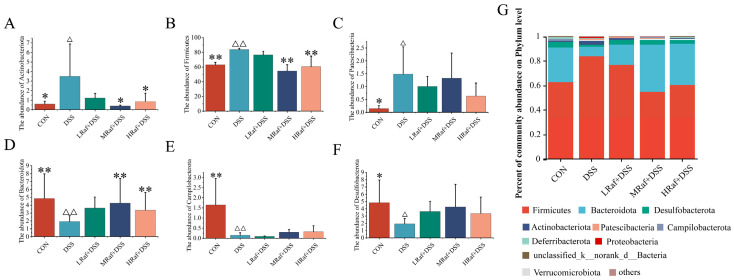
Effects of Raf on gut microbial composition at the phylum level. Abundances of (**A**) *Actinbacteriota*, (**B**) *Firmicutes*, (**C**) *Patescibacteria*, (**D**) *Bacteroidota*, (**E**) *Campilobacterota*, and (**F**) *Desulfobacterota*. (**G**) Role of Raf in regulating gut microbiota structure at the phylum level. ^Δ^
*p* < 0.05, ^ΔΔ^
*p* < 0.01, relative to the control. * *p* < 0.05, ** *p* < 0.01, relative to the DSS group. CON: control; Raf: Raffinose; DSS: dextran sulfate sodium salt.

**Figure 9 foods-13-01849-f009:**
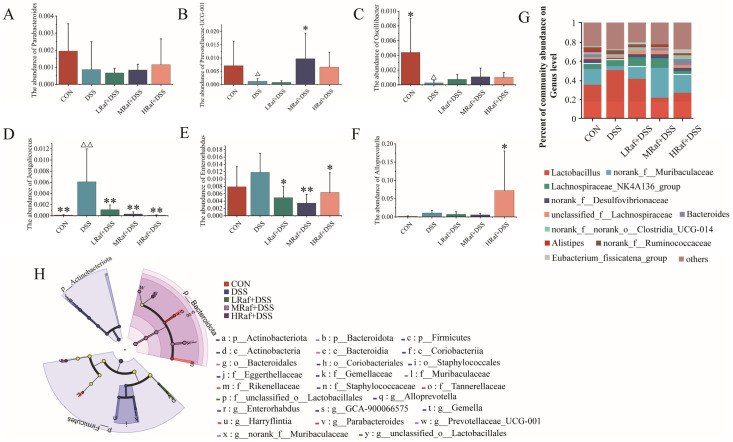
Effects of Raf on gut microbial composition at the genus level. Abundances of (**A**) *Parabacteroides*, (**B**) *Prevotellacceae-UCG-001*, (**C**) *Oscillibacter*, (**D**) *Jeotgalicoccus*, (**E**) *Enterorhabdus*, and (**F**) *Alloprevotella*. (**G**) Role of Raf in regulating gut microbiota structure, at the genus level. (**H**) LEfSe analysis. ^Δ^
*p* < 0.05, ^ΔΔ^
*p* < 0.01, relative to the control. * *p* < 0.05, ** *p* < 0.01, relative to the DSS group. CON: control; Raf: Raffinose; DSS: dextran sulfate sodium salt.

**Figure 10 foods-13-01849-f010:**
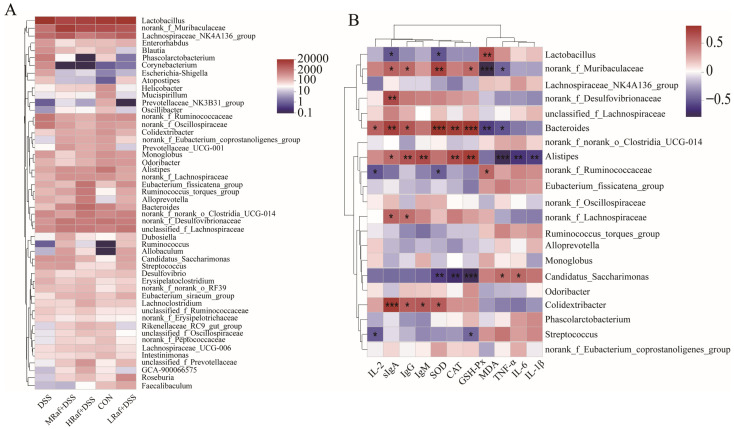
Heatmap of the correlation analysis results. (**A**) Correlation analysis between the treatment groups and microbiotic genus abundances. (**B**) Correlation analysis between oxidative stress, inflammation, and immune-related cytokine levels and microbiotic genus abundances. CON: control; Raf: Raffinose; DSS: dextran sulfate sodium salt. * *p* < 0.05, ** *p* < 0.01, and *** *p* < 0.001.

**Figure 11 foods-13-01849-f011:**
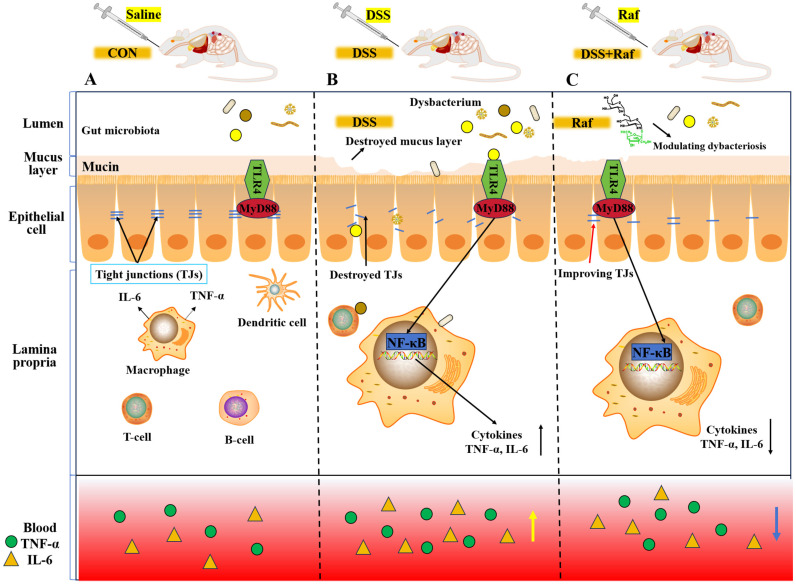
Mechanisms underlying the inhibitory effect of Raf on DSS-induced colitis. (**A**) Healthy control. (**B**) DSS-induced colitis model. (**C**) Raf-pretreated colitis group. Raf significantly regulated DSS-induced dysbiosis, restoring it to a normal level. Raf downregulated the protein and gene expression of *TLR4* and *MyD88*, thus blocking the *TLR4–MyD888–NF-κB* signaling pathway and thereby reducing the DSS-induced elevated levels of colitis-associated inflammatory cytokines. Lamina propria: Upward black arrows indicate an increase in content, downward black arrows indicate a decrease in content; Blood: Upward yellow arrows indicate an increase in content, downward blue arrows indicate a decrease in content.

**Table 1 foods-13-01849-t001:** Primers sequence for qRT-PCR.

Gene Name	Forward(5′-3′)	Reverse(5′-3′)
*β-actin*	ATATCGCTGCGCTGGTCG	GATCTTCTCCATGTCGTCCC
*TLR4*	GCACTGTTCTTCTCCTGCCT	AGAGGTGGTGTAAGCCATGC
*MyD88*	GCATGGTGGTGGTTGTTTCTG	GAATCAGTCGCTTCTGTTGG
*NF-κB*	ACACTGGAAGCACGGATGAC	TGTCTGTGAGTTGCCGGTCT
*TNF-α*	CAAAATTCGAGTGACAAGCC	TTGTCCCTTGAAGAGAACCT
*IL-6*	GGGACCCGAGTTACTACTTG	CTGGGCTCTGCTATCCAAGG

**Table 2 foods-13-01849-t002:** Effect of Raf on body weight and the thymus and spleen index in DSS-induced colitis model mice.

Group	Raf	Initial Weight (g)	Final Weight (g)	Daily Weight Gain (g)	Thymus Index (mg 10 g^−1^)	Spleen Index (mg 10 g^−1^)
CON	—	20.83 ± 1.18	33.89 ± 0.87 **	0.55 ± 0.05 **	34.72 ± 2.68 **	33.95 ± 1.71 **
DSS	—	20.71 ± 0.99	26.48 ± 1.76 ^ΔΔ^	0.24 ± 0.09 ^ΔΔ^	18.45 ± 3.22 ^ΔΔ^	20.93 ± 4.19 ^ΔΔ^
LRaf+DSS	100	20.63 ± 1.25	29.81 ± 1.09 **	0.38 ± 0.06 **	21.87 ± 3.03	22.56 ± 3.73
MRaf+DSS	200	20.94 ± 0.84	30.74 ± 0.88 **	0.41 ± 0.06 **	26.48 ± 3.92 **	26.05 ± 4.02 **
HRaf+DSS	400	20.81 ± 0.79	32.19 ± 0.76 **	0.47 ± 0.07 **	29.91 ± 2.33 **	29.15 ± 1.95 **

Data are represented as mean ± SE, n = 8. ^ΔΔ^ *p* < 0.01, relative to the control. ** *p* < 0.01, relative to the DSS group. CON, control; DSS: dextran sulfate sodium salt.

## Data Availability

The datasets presented in this study can be found in online repositories. The names of the repository/repositories and accession number(s) can be found at: PRJNA1100313.

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
