# Peer review of "Raffinose Ameliorates DSS-Induced Colitis in Mice by Modulating Gut Microbiota and Targeting the Inflammatory TLR4–MyD88–NF-κB Signaling Pathway"

_foods, 2024, doi:10.3390/foods13121849_

Round 1
Reviewer 1 Report
Comments and Suggestions for Authors
The subject entitled: The alleviation of colitis by DSS-induced though modulation of gut microbiota and inflammation-related TLR4-MyD88-NF- κB signaling pathways by Raffinose in mice submitted for our evaluation is very interesting because it has an obvious and particular scientific interest. In doing so, we will also make our scientific contribution to qualitatively improve the scientific value of the manuscript article.
Introduction is based on a scientific approach, the authors talk about colitis, its corollaries etc., but unfortunately they don't sufficiently address the problem with epidemiological data. So it's not clear on what basis the authors focused on colitis. Others are not clearly defined the problem. As a result, the introduction needs to be further improved.
Methods section: In figure 1 of the experimental animal diagram, the authors have chosen doses of 100 mg/kg Raf, 200 and 400 mg/kg Raf. On what basis were these choices made? The authors should reference the method or justified the choices of these doses.
Results section: The authors should improve the quality of the figure 9.
The discussion lacks arguments to better illustrate the results. As a result, discussion must be improved.
English needs to be improved as the use of certain expressions leads to confusion.

English needs to be improved
Reviewer 2 Report
Comments and Suggestions for Authors
The study presents interesting findings of raffinose application to alleviate DSS-induced colitis.
Some parts should be improved/corrected:
1. Please indicate the novelty of the study.
2. Line 19 abstract - editorial error
3. Indroduction first sentence, please rephrase - "is a trisaccharide" and "is non-digestible"
4. Line 37 - a full name of E. coli should be given - please correct
5. Line 60 - please use italics for in ovo
6. Table 2 - please use superscript for "-1"
7. Line 433 - immune instead of "im-mune"
Comments on the Quality of English Language
Minor editing of English language required
Reviewer 3 Report
Comments and Suggestions for Authors
Raffinose is an oligosaccharide that acts as a prebiotic and is fermented by the intestinal microbiota, which contributes to controlling the microbial balance, which in turn helps prevent colitis. The study examined the potential effect of raffinose in reducing ulcerative colitis induced by dextran. Efforts have been made in the study, but many points must be taken into consideration for the work to be complete.
l The title needs linguistic checking.
l The current status of the abstract section is a large introduction, some results, and at the end, it was concluded that "raffinose was the promising prebiotics in animal intestinal health". It is better to re-edit the abstract section so that it summarizes in a balanced manner the rationale for the study, the aim of the study, the methods, the results, and finally a conclusion that answers the main question of the study: whether the potential effect of raffinose in preventing ulcers has been achieved or not.
l Please remove the last paragraph in the introduction section, about the results obtained, and add instead the objectives of the study.
l In the Materials and Methods section, please consider the following points:
- Check the subheadings because they are duplicates.
- Please add the reference to the design of the experiment and mention the basis used to determine the doses.
- In line 119, the authors mentioned that they extracted the liver and spleen and weighed them, and in the following parts were the spleen and thymus. Please check.
- The organ index is calculated by the formula: Organ index (%) = weight of organ/weight of body x 100. please check.
- It is best to calculate the weight gain and add the results to Table 2 so that it is easy to identify the effect of dextran and raffinose on weight.
- It is best to calculate the disease activity index (DAI) so that it is easy to identify the effect of dextran and raffinose on the symptoms of ulcers, such as weight loss, diarrhea, and bloody stool.
l I noticed some references in the results section, and it would be better to combine the results and discussion parts, keeping in mind that the discussion would be better if it was based on the interpretation of the results and their comparison with other studies, for example, on the effect of dextran and studies on the anti-ulcer effect of raffinose-like sugars such as stachyose, which has one galactose unit over raffinose, that was studied in the Reference No. 34 mentioned by the authors.
There are also studies on the anti-ulcer effect of some foods containing raffinose, such as chickpeas, and studies on the effect of raffinose on the gut microbiota, all of which could be used to write a good discussion section.

The manuscript needs good linguistic checking.
Round 2
Reviewer 2 Report
Comments and Suggestions for Authors
The authors revised the manuscript as suggested.
Reviewer 3 Report
Comments and Suggestions for Authors
The suggested modifications were made and the manuscript became suitable for publication.